# Development of Scales to Measure and Analyse the Relationship of Safety Consciousness and Safety Citizenship Behaviour of Construction Workers: An Empirical Study in China

**DOI:** 10.3390/ijerph16081411

**Published:** 2019-04-19

**Authors:** Xiangcheng Meng, Huaiyuan Zhai, Alan H. S. Chan

**Affiliations:** 1School of System Engineering and Engineering Management, City University of Hong Kong, Hong Kong, China; xcmeng3-c@my.cityu.edu.hk (X.M.); alan.chan@cityu.edu.hk (A.H.S.C.); 2School of Economics and Engineering Management, Beijing Jiaotong University, Beijing 100044, China

**Keywords:** construction workers, safety consciousness, safety citizenship behaviour, questionnaire survey

## Abstract

China’s construction industry has experienced a long period of development and reform but compared to developed countries, safety on construction sites in China continues to present serious problems. Safety consciousness and safety citizenship behaviour are influential factors related to safety issues in the construction industry and may play a direct role in improving the safety of personnel on construction sites. However, recently no research has been focused on the relationship between safety consciousness and safety citizenship behaviour. Therefore, this paper aimed to investigate the relationship between safety consciousness and safety citizenship behaviour for personnel working on construction sites in China by using a questionnaire survey and statistical analysis, so that correlation between safety consciousness and safety citizenship can be demonstrated and effective measures suggested to improve the safety of construction workers in China, and perhaps in other countries as well.

## 1. Introduction

China’s construction industry has developed rapidly since 2012 due to continuous development of the economy. However, with the great increase in construction activities and projects, accidents and injuries have become serious issues in the construction industry. Given the low profit margin compared with other industries, there has been a lack of construction-related research on safety, which has led to a higher accident rate in the construction industry compared with other industries. According to the global statistics of the International Labour Organisation [1], the accident rate in the construction industry was three times higher than that of other industries and the mortality rate of construction workers was five times that of all industries. According to Meng et al. [2], the data, which were collected between 2010 and 2016, revealed that 3817 fatal accidents occurred during the construction of buildings and municipal facilities in China. Also, according to the Ministry of Housing and Urban-Rural Development of China [3], in 2017, 692 accidents occurred, and 807 construction workers died in China. Compared with 2016, the number of accidents in 2017 increased by 58, which means an 8.38% year-on-year growth. These numbers underline the extreme importance and urgency of reducing construction accidents and improving the safety performance of construction personnel.

Causes of accidents are commonly grouped into three categories: technical failures, management issues, and human factors. Considerable research attention has been focused on reducing human factors related safety problems and several studies have attributed the frequent occurrence of accidents to weak safety consciousness among workers. Fang et al. [4], for example, pointed out that work performance and efficiency can be enhanced by improving the safety consciousness of construction workers. It has also been pointed out that safety citizenship behaviour can affect safety performance. Lingard et al. [5] demonstrated that co-workers increase the risk to each other if they are unable to abide by correct safety citizenship behaviour, thereby leading to dangerous situations. However, recently no studies have focused on verifying the relationship between safety consciousness and safety citizenship behaviour. If a relationship between safety consciousness and safety citizenship behaviour can be demonstrated, the link between these two constructs will become clear, which can have a positive effect on occupational safety and health improvement for construction workers. Therefore, this paper aimed to develop effective questionnaire-based measurement scales to be used to verify the relationship between safety consciousness and safety citizenship behaviour for workers on construction sites by conducting a survey using the developed questionnaire and subsequent correlation analysis. The results of the data analysis were then to be used to provide recommendation to improve the safety of workers in the construction industry.

## 2. Literature Review

### 2.1. Safety Consciousness

Safety consciousness is defined as the perception and understanding of safety with regard to environment and circumstance. It has been suggested that the level of safety can be improved through promotion of safety consciousness [6,7,8,9]. Chan et al. [10] found a strong correlation between safety consciousness and the safety performance of construction personnel and explored how construction safety may be improved in mainland China and Hong Kong by using a survey of safety attitudes.

The factors related to safety consciousness that have been studied quite widely are safety regulations and safety education. Zhang and An [11] found a direct relationship between worker understanding of safety procedures and level of safety consciousness. De Koster et al. [12] also showed that when construction personnel are familiar with the relevant laws and regulations, their thinking is influenced, helping to improve their safety consciousness levels. Bradford et al. [13] found that training and education on safety skills can positively influence the awareness of workers to avoid risks. Hinze and Gambatese [14] and Misiurek et al. [15] found that popularising safety knowledge among workers and carrying out safety education activities cultivates and promotes safety consciousness, therefore improving the occupational safety of construction workers.

The effect of conscientiousness has also been discussed in the literature. According to Roth and Brooks-Gunn [16], conscientiousness, which is related to risk prevention ability, can be defined as whether people still follow safety rules in the absence of supervision. Dudley et al. [17] studied the influence of conscientiousness on safety consciousness and found that conscientiousness was related positively to self-efficacy, which was related positively to safety performance. Others have examined the influence of work experience on safety consciousness. For instance, according to Siu et al. [18], experienced older workers have decreased risk at work. They also point out that due to fewer job opportunities for older workers they may be more willing than younger workers to follow safety rules. However, Chen and Wang [19] report that experienced workers tend to solve problems quickly using their existing work experience and ignore considerations of their own safety status and the related regulations. Therefore, the experience level of construction workers can be both drivers and constraints for enhancing productivity (Javed et al. [20]).

### 2.2. Safety Citizenship Behaviour

For safety citizenship behaviour, Hofmann and Morgeson [21] proposed a clear concept of safety citizenship behaviour, which was defined as a voluntary personal behaviour produced by construction personnel in order to ensure the safety performance of other team members and achieve the safety performance of the project and organisation. They found that the concept of safety citizenship behaviour was important for improving the safety performance of working groups and emphasising mutual support between employees and could improve organisational effectiveness. Shama et al. [22] gave further clarifications of the concept of safety citizenship behaviour, defining it as voluntary assistance to other project members and project organisations to achieve safety improvements and working conditions. Conchie and Donald [23] found that safety citizenship behaviour, as a specific organisational citizenship behaviour, included acts to protect the safety of other people, endeavouring to prevent the occurrence of accidents and proactively striving to improve organisational safety systems and general conditions of safety in the workplace.

Recently, there has been increasing research interest in uncovering factors relating to safety citizenship behaviours. According to Du and Zhao [24], mutual help between colleagues was classified as one element of safety citizenship behaviour of mining workers. Turner et al. [25] reported that the safety citizenship behaviour of employees increased when workers felt mutual concern and care for each other. Curcuruto and Griffin [26] found that organisational support, which is a factor of safety citizenship behaviour, was influenced by the extent of mutual help between construction workers. Also, an influence on safety citizenship behaviour was found for the quality of vertical relationships among construction working groups. Gerstner and Day [27] concluded that the leader-member exchange (LMX), which refers to the relationship between superior and employee, has a direct and positive impact on workers’ organisational commitment and behaviour. Reader et al. [28] demonstrated that higher quality of social exchange relations between superior and subordinate can positively influence organisational support, thereby improving worker safety citizenship behaviour.

Safety citizenship behaviour can also be enhanced by worker’s frequent participation in safety suggestion and communication. According to Turner et al. [25], making suggestions and expressing opinions about safety matters contributed to the measurement of safety citizenship behaviour, thus achieving promotion and development of safety for the whole organisation. In addition, Sharon [29] found that frequent participation in making suggestions can significantly increase an employee’s sense of belonging and make the organisation stronger. It is essential that suggestions are taken seriously and treated fairly. The impact of self-control has also been considered and Heatherton and Baumeister [30] defined self-control as encompassing a wide range of responses including the ability to avoid potential risks instinctively. According to Hagger et al. [31], self-control was related to initiative in safety participation, and was as key factor when studying organisational behaviour across national groups.

### 2.3. Dimensions

To study the correlation between safety consciousness and safety citizenship behaviour, the two constructs must be quantitatively measured. Based on the literature review above, safety consciousness and safety citizenship behaviour can each be measured through four related dimensions. Therefore, the initial version of the two measurement scales covered the corresponding four dimensions, as shown in Table 1, and the questionnaire items were designed based on each dimension (3 items for each dimension). Revisions need to be considered, if necessary, due to designed item inaccuracy or overlapping factors, so as to improve these newly developed scales. The details of scale development will be discussed in the next section.

### 2.4. Research Hypotheses

After defining all the dimensions, the relationship hypothesis between safety consciousness and safety citizenship behaviour can be established and verified. Also, the hypotheses about the correlations between the dimensions can be established so that suggestions can be put forward for the particular dimensions to provide more detail than is the case at present using the whole abstract construct. In this study, nine hypotheses were put forward for construction workers (see Table 2).

## 3. Methodology

### 3.1. Development of Scales

The literature review revealed two important research issues, namely, safety consciousness and safety citizenship behaviour. However, at present there are no measurement scales for safety consciousness and safety citizenship behaviour in the construction industry. Up until now, a measurement scale for safety consciousness has been used in the agriculture, driving, and catering industries [9,12,32], while a safety citizenship behaviour scale was developed for the manufacturing industry by Hofmann and Morgeson [21]. For the study reported here, the particular original version of scales and items for safety consciousness and safety citizenship behaviour was identified through an extensive review of publications, safety specifications, and construction guidance for the construction industry. The initial structures for both scales covered four dimensions, but with different questionnaire items for each dimension. There were 24 items in total, each measured using a 5-level Likert Scale. The details for the two scales are shown in Appendix A. Statistical analysis, such as exploratory factor analysis (EFA), is needed to test the effectiveness of the scales, so that any inappropriate item and dimension can be removed or revised based on the results of data analysis. Details about data analysis are provided in the following section.

### 3.2. Questionnaire Survey

A questionnaire survey was conducted via an online platform, aimed at data collection and statistical analysis. Four hundred construction workers were asked to fill out a specially-designed questionnaire, which was segmented into three main sections. The reasons for choosing online questionnaire distribution are discussed by Wright [33] and Seki et al. [34]. Online questionnaire surveys have a number of advantages compared with field distribution, such as cost, time saving and access to unique populations. The purpose of the survey was explained to participants in the first section before they filled in the questionnaire. After that, the main types of questions were introduced and then the demographic information (gender, educational background, age, and weekly working hours) for each participant was collected. The full version of the questionnaire is given in Appendix A. The survey obtained 382 valid answers after examining all of the data collected and represented a valid response rate of 95.5%. The non-valid responses were mainly unfinished questionnaires, and these were not used in further data analysis.

### 3.3. Data Analysis

Data analysis was conducted after data collection and collation were completed. SPSS 24.0 (IBM, Armonk, New York, NY, USA) and AMOS 24.0 (IBM, Armonk, New York, NY, USA) were used for data processing and statistical analysis. For data processing, exploratory factor analysis (EFA) was conducted first to extract and synthesise the overlapping parts of the original variables into factors so that the initial scale versions for both safety consciousness and safety citizenship behaviour could be revised, then the reliability and validity tests of the scales were conducted by Cronbach Coefficient and confirmatory factor analysis to ensure their effectiveness. The hypotheses were then tested using Pearson correlation analysis and structural equation modelling technique. In addition, linear regression models for both single element and multiple elements were also established. Analysis of variance (ANOVA) was applied to identify the significant differences in safety citizenship behaviour between groups with different personal characteristics. Specific improvements were considered based on the results of the data analysis.

## 4. Results

### 4.1. Factor Analysis for Scale Development

According to Clark and Watson [35], the primary measure of scale development is exploratory factor analysis, which is used to extract and synthesise the overlapping parts of the original variables into factors within the scale. If the original variables are independent of each other, then the degree of correlation is very low. If information overlap exists, then no common factor exists, and hence, no factor analysis is needed. Therefore, before factor analysis, Kaiser-Meyer-Oklin Measure of Sampling Adequacy (KMO) and the Bartlett test of sphericity are used to analyse whether the original variables correlate or are suitable for factor analysis.

Table 3 and Table 4 showed that KMOs for both parts (safety consciousness and safety citizenship behaviour) of the questionnaire were higher than 0.6, which indicated the feasibility of further exploratory factor analysis. After the rotation of matrices, nine factors were obtained (Table 5 and Table 6). According to the results of exploratory factor analysis, there were nine underlying factors within the two scales. However, as the factor loading for factor 5 was low, this factor was removed by deleting the corresponding item 9, so that the number of items measuring factor 2 became two. Results of the rotational matrices for other items were acceptable because the items measuring the same factor (dimension) had strong correlations between them. The remaining eight factors all reflected one of the dimensions of the initial scale version: familiarity with safety regulations (Factor 1), training and education of safety skills (Factor 2), conscientiousness (Factor 3), dependency level of work experience (Factor 4), mutual aid among the workers (Factor 6), relationship between superior and subordinate (Factor 7), participation of suggestion making (Factor 8), and self-control (Factor 9).

### 4.2. Test of Reliability and Validity

The reliability and validity of the scales were tested to demonstrate their effectiveness. Cronbach coefficient was used to measure the reliability of the questionnaire [36,37]. A Cronbach’s alpha value above 0.70 is recommended to ensure data reliability [38]. The results of data analysis (see Table 7) showed that reliabilities of the safety consciousness scale and the safety citizenship behaviour scale were satisfied, and the items for both safety consciousness and safety citizenship behaviour had good reliability, namely, 0.751 for those of safety consciousness and 0.813 for safety citizenship behaviour.

For the validity test, after the exploratory factor analysis (EFA) was conducted and all dimensions for both safety consciousness and safety citizenship behaviour were classified, and confirmatory factor analysis (CFA) was conducted to further verify the validity of both portions of the questionnaire. Pintrich et al. [39] described confirmatory factor analysis as a statistical analysis for social survey data. It tests whether the relationship between a factor and the corresponding measured items conform to the theoretical relationship designed by the researchers. According to van Prooijen and van Kloot [40], confirmatory factor analysis is often tested by structural equation modelling. Given the practical scientific research of Anna and Jason [41] and Schreiber et al. [42], CFA is used to test measurement models by using test coefficients, such as Average Variance Extracted (AVE). In this study, 180 construction workers were recruited to redo the revised questionnaire survey and the collected data was used for CFA. Among all the test coefficients, χ^2^/df directly checks the similarity between the sample covariance matrix and the estimated variance matrix. Root Mean Square Residual (RMR) is applied to measure the average residual of correlation between prediction and observation. The Goodness-of-Fit Index (GFI) refers to the degree of fitting of the regression line to the observed values. The Incremental Fitness Index (IFI) and Comparative Fitness Index (CFI) are also applied to measure the fitness of the regression. In addition, the Root Mean Square Error of Approximation (RMSEA) is applied for the fitness test. Table 8 shows that apart from some minor variances, most corresponding criteria were satisfied when compared with statistical standards. Therefore, both portions of the questionnaire had good validity.

### 4.3. Correlation Analysis

After testing the reliability and validity of the questionnaire, the scores for safety consciousness and safety citizenship behaviour for all participants were summarised and averaged, then a Pearson correlation test was conducted using the average scores of safety consciousness and safety citizenship behaviour to test the authenticity of the hypotheses (Francisco et al. [43]). Results indicate that at the 0.01 level (double tail), the Pearson correlation between safety consciousness and safety citizenship behaviour was 0.602, which presents a positive relationship between those two concepts. Thus, H1 was verified and accepted.

To verify the authenticity of hypotheses H2–H9, correlation analysis was carried out on each dimension for safety consciousness and safety citizenship behaviour. The results (Table 9 and Table 10) show that apart from H5, which yielded insignificant correlations, all the other hypotheses were accepted at the 0.01 level. The strongest correlation was between “Safety skills training” and “Safety citizenship behaviour”, followed by “Familiarity with safety regulations”. Thus, safety skills training can be combined with safety regulations and laws to increase the performance of the safety citizenship behaviour of construction workers. More detailed discussion of these results is provided below.

Correlation results between each dimension of safety consciousness and safety citizenship behaviour are shown in Table 11. All dimensions of safety consciousness are significantly correlated with the dimensions of safety citizenship behaviour and with the relationship between superior and subordinate. However, there were low correlations between “the relationships between superior and subordinate” and the three dimensions of safety consciousness, namely, safety skills training, dependency on experience, and conscientiousness.

In addition, the structural equation model (SEM) was also established to conduct the path coefficient analysis. The structural details of the model are depicted in Figure 1 and the fitness indices are shown in Table 12. All the eight dimensions in Table 12 were abbreviated: the “SC” is “safety consciousness”, “SCB” is “safety citizenship behaviour”, “regulation” is “Familiarity with safety regulations”, “education” is “Training and education of safety skills”, and “experience” is “Dependency level of work experience”. For the safety citizenship behaviour, “help” is “Mutual aid among the workers”, “relation” is “Relationship between superior and subordinate”, and “suggestion” is “Participation in suggestion making”. Also, the residual variables were set for each indicator variable (rectangular ones) and the endogenous latent variable (SCB) from e1 to e9, thus describing the part of an endogenous variable which cannot be explained.

According to the criterion of the fitness indices shown in Table 12, each index satisfied the appropriate level, which indicated that the overall fitness of the initial model was good. Also, as shown in Table 13, the path coefficient of the initial theoretical model, specifically, the path coefficient of the relationship between safety consciousness (SC) and safety citizenship behaviour (SCB) was 0.834, which indicated that safety consciousness positively influenced safety citizenship behaviour. The coefficient of other paths and their corresponding significances are also given in Table 13.

### 4.4. Regression Model Establishment

According to the above section, a positive correlation between safety consciousness and safety citizenship behaviour was established, and here, further analysis was performed to determine whether or not the relationship is linear. Both single and multiple linear regression models were established, and the single linear model is shown in Figure 2. The *R*^2^ value was 0.663, which verified that the significance of the correlation between the two concepts. The equation for the single linear regression model is expressed below.
Safety Citizenship Behaviour = 1.87 + 0.57 × Safety Consciousness(1)

For the linear regression model with multiple elements, safety citizenship behaviour was taken as the dependent variable. All four dimensions of safety consciousness, namely, safety skills training, dependency on experience, conscientiousness and familiarity with the laws and regulations, were adopted as the multiple elements. The results of the simulation are shown in Table 14.

According to the tolerance and variance inflation factor (VIF) test, there were no multiple collinear problems among the variables. The adjusted *R*^2^ was 0.681, indicating that the total of all the variables explain 68.1% of the variance of safety citizenship behaviour. With the exception of dependency of experience, all the corresponding *p* values were lower than 0.05, therefore, safety skills training, conscientiousness, and familiarity with the laws had a significant positive influence on safety citizenship behaviour. The regression equation is expressed as:Safety Citizenship Behaviour = 1.436 + 0.084 × Safety Skills Training + 0.338 × Conscientiousness + 0.260 × Familiarity with the laws(2)

### 4.5. Subgroup ANOVA

Based on the collected demographic information, different groups were defined in terms of personal data, as shown in Table 15. Analysis of Variance (ANOVA) was then used to analyse subgroup differences to identify the particular samples which need to be given extra attention [44].

The results showed that there were significant subgroup differences in different genders, educational backgrounds, and weekly working hours. Therefore, specific suggestions will be provided to target those particular groups of workers and will be discussed in the next section.

## 5. Discussion

The Pearson correlation test showed that safety skills training was strongly correlated with safety citizenship behaviour, therefore, extra emphasis should be given to this particular dimension. A popular way of conducting safety training is by recalling and analysing accident scenes to provide relevant direct experience. According to Eiris et al. [45] and Wang et al. [46], safety management based on visualisation technology is a promising method for improving the safety of construction workers. It can be used to provide a virtual simulation of accident sites so that construction workers can readily understand the causes of accidents and the related prevention measures. As indicated by the results of Pearson correlation analysis, safety citizenship behaviour was also shown to be mostly related to safety regulations and laws compared with other dimensions of safety consciousness. Therefore, conducting safety education in combination with learning safety laws and regulations may significantly strengthen construction worker’s familiarity with safety rules. According to Matt et al. [47], adopting workplace safety regulations will lead to a significant reduction in accidents and worker injuries. Also, Nielsen [48] found that safety regulations can promote a good safety climate, thus improving the safety performance of construction workers. In addition, process evaluation can be conducted to test learning outcomes, and therefore ensure the quality of safety training and education. According to Ken [49] and Riedel et al. [50], appropriate rewards and punishments can also be used to promote learning initiatives and effect.

Also, the subgroup ANOVA found significant differences in safety citizenship behaviour among genders, educational backgrounds, and weekly working hours, and specific suggestions are provided here for those subgroups.

For different genders (Table 16), the safety citizenship behaviour of female workers was worse than male workers. The most likely reason mentioned by Jacqueline [51] is that male workers are generally influenced by Chinese traditional culture associated with personal loyalty and the macho imagination. They get on well with co-workers more easily, thereby contributing more to mutual assistance and organisational safety. To improve the safety citizenship of female workers, the job satisfaction and safety performance of female workers need to be emphasised and measures need to be established to improve their responsibility for the safety of others and the support of both supervisors and male co-workers [52]. Sexual harassment and gender discrimination should be prohibited in the workplace and firmly handled by management and supervisors to encourage the integration and participation of female workers into the workforce [53]. Also, related analyses coupled with training and compensatory strategies for better decision making should be used to reduce all possible negative influences on the performance of female workers [54].

Also, workers with low levels of educational background may have poorer understanding of safety-related knowledge and theory due to their relatively short exposure to education, therefore their safety knowledge and risk prevention abilities may not be adequate [55]. The safety citizenship behaviour of construction workers was better for those with more education. Therefore, paying special attention to the safety education of employees with a low educational background is imperative if improvements in safety management are to be achieved [56].

This study also considered the effects of weekly working hours. Dembe [57] and Skogstad et al. [58] found that unreasonable work hours had a strong effect in leading to increased occupational injuries and illnesses. Alameddine et al. [59] found that unsuitable work hours can negatively affect work productivity, job satisfaction, worker health, and inconsistent job performance. Bowen et al. [60] discussed the negative effect of imbalances in working-life. The results shown in Table 15 suggest that reasonable weekly working time for construction workers should be set between 46 and 50 h, when the corresponding average score for safety citizenship behaviour was highest. However, the analysis of this part was not detailed enough, and further study should be conducted to determine comprehensive recommendations for working hours for the various practical daily and weekly work situations experienced by construction workers.

## 6. Conclusions

This study aimed to develop reliable measurement scales and to analyse the relationship between safety consciousness and safety citizenship behaviour. Data from 382 Chinese construction workers was collected by means of a questionnaire survey. The scales were developed after conducting an extensive literature review and exploratory factor analysis (EFA), as a result of which both scales were divided into four dimensions. The reliability and validity were tested by Cronbach’s alphas and CFA. The correlation analysis indicated that safety consciousness was positively related to safety citizenship behaviour and both simple and multiple linear regression models were established to further analyse the correlation between those two concepts. In terms of the defined dimensions, it was found that three dimensions of safety consciousness significantly influenced safety citizenship behaviour and three dimensions of safety citizenship behaviour were influenced by increasing the extent of safety consciousness. Attention should be directed towards education and training as a top priority because the highest correlations found here were between safety education and safety citizenship behaviour compared with other dimensions of safety consciousness. Also, subgroup ANOVA was conducted in terms of different personal characteristics, thus providing knowledge to more effectively improve the safety status of construction workers in a more targeted way. Results here also showed that low levels of safety citizenship behaviour were mainly among female workers, poorly-educated workers, and overworked workers of both genders. Particular attention needs to be given to these three groups of workers. Also, reasonable working hours must be established to avoid low safety performance due to long working hours.

However, this study had some limitations. First, demographic information did not include different types of construction work, so the subgroup ANOVA was not applied to workers with different crafts, such as cementer, crane operator, and concrete-reinforcement worker. For further research, the questionnaire items may need to be redesigned so that the information for different types of crafts can be collected. Second, in this study, the data obtained for relationship analysis were mainly collected from self-reporting measures (questionnaire survey), which may result in not entirely accurate responses. To solve this problem, several negatively-keyed items were designed in this study, though the balance of positive and negative keyed items was not perfect [61]. In addition, recall and response bias may also cause inaccuracy of the results. However, Cronbach’s alphas for all factors indicated good reliability for the measures used. Nevertheless, multiple data sources are recommended for future research to reduce any possible problems caused by self-reports.

## Figures and Tables

**Figure 1 ijerph-16-01411-f001:**
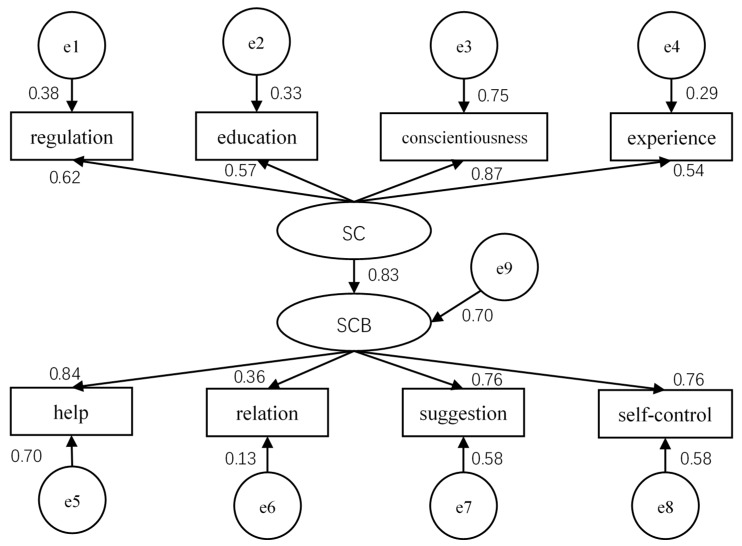
Structural equation model of safety consciousness and safety citizenship behaviour.

**Figure 2 ijerph-16-01411-f002:**
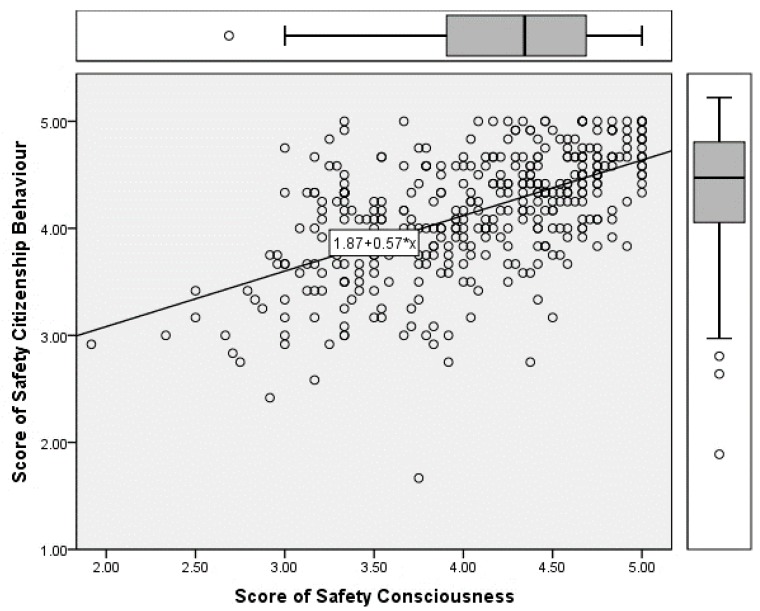
Single linear model for Safety Citizenship Behaviour and Safety Consciousness. The *R*^2^ value was 0.663, which confirmed the significant linear correlation between the variables.

**Table 1 ijerph-16-01411-t001:** Dimensions of safety consciousness and safety citizenship behaviour in the initial version

Construct	Dimensions	References
Safety consciousness	Familiarity with safety regulations	[11,12]
	Training and education of safety skills	[13,14,15]
	Conscientiousness	[16,17]
	Dependency level of work experience	[18,19,20]
Safety citizenship behaviour	Mutual aid among the workers	[24,25,26]
	Relationship between superior and subordinate	[27,28]
	Participation of suggestion making	[29,30]
	Self-control	[31,32]

**Table 2 ijerph-16-01411-t002:** Research hypotheses for the relationship between safety consciousness and safety citizenship behaviour for construction workers.

Number	Content
H1	Significant positive correlations exist between safety consciousness and safety citizenship behaviour.
H2	The greater the familiarity with safety regulations, the higher the level of safety citizenship behaviour.
H3	The greater the attention paid to safety skills training, the higher the level of safety citizenship behaviour.
H4	The stronger the conscientiousness, the higher the level of safety citizenship behaviour.
H5	The higher the dependency on construction experience, the lower the level of safety citizenship behaviour.
H6	The influence of safety consciousness on safety citizenship behaviour is reflected in the extent of workers’ mutual aid.
H7	The influence of safety consciousness on safety citizenship behaviour is reflected in the feedback relationship between superior and subordinate.
H8	The influence of safety consciousness on safety citizenship behaviour is reflected in the participation of suggestion making.
H9	The influence of safety consciousness on safety citizenship behaviour is reflected in self-control.

**Table 3 ijerph-16-01411-t003:** KMO (Kaiser-Meyer-Oklin) and Bartlett Test of Sphericity for the data of safety consciousness.

KMO and Bartlett Test of Sphericity		
KMO measure of sampling adequacy		0.795
	Approximate chi-square	1244.674
	Freedom	55
	Significant	0.000

**Table 4 ijerph-16-01411-t004:** KMO and Bartlett Test of Sphericity for the data of safety citizenship behaviour.

KMO and Bartlett Test of Sphericity		
KMO measure of sampling adequacy		0.889
	Approximate chi-square	1740.348
	Freedom	66
	Significant	0.000

**Table 5 ijerph-16-01411-t005:** Rotational matrix of safety consciousness.

	The Component Matrix After Rotation					
	Items	Factor 1	Factor 2	Factor 3	Factor 4	Factor 5
Factor 1	Q1	0.850	0.172	−0.041	0.131	−0.148
	Q3	0.808	−0.023	−0.091	0.112	0.216
	Q2	0.805	−0.212	0.182	0.214	0.171
Factor 2	Q5	0.225	0.826	0.244	0.193	−0.271
	Q4	0.137	0.783	0.031	−0.234	−0.335
Factor 3	Q6	−0.278	0.134	0.839	−0.355	0.126
	Q7	−0.303	−0.312	0.772	−0.231	0.376
	Q8	−0.135	0.215	0.645	0.134	−0.398
Factor 4	Q11	0.032	0.251	0.012	0.825	−0.315
	Q10	−0.217	−0.182	0.011	0.628	−0.012
	Q12	−0.104	−0.083	−0.082	0.533	0.231
Factor 5	Q9	−0.211	0.216	0.218	0.091	0.571

**Table 6 ijerph-16-01411-t006:** Rotational matrix of safety citizenship behaviour.

	The Component Matrix After Rotation				
	Items	Factor 6	Factor 7	Factor 8	Factor 9
Factor 6	X1	0.788	0.032	−0.145	0.268
	X2	0.775	0.122	0.027	−0.273
	X3	0.756	−0.117	−0.213	0.083
Factor 7	X6	0.192	0.803	0.341	0.034
	X4	−0.162	0.729	−0.213	0.193
	X5	0.013	0.707	−0.124	0.117
Factor 8	X7	−0.214	0.341	0.748	0.022
	X8	−0.301	−0.112	0.723	0.128
	X9	0.277	−0.362	0.614	−0.214
Factor 9	X12	0.371	−0.231	0.034	0.867
	X10	0.246	0.341	0.215	0.662
	X11	−0.012	0.012	0.211	0.645

**Table 7 ijerph-16-01411-t007:** Overall reliability of the questionnaire.

Portion	Dimensions	Cronbach Coefficient	Total Cronbach Coefficient	Number of Items
Safety consciousness	Familiarity with safety regulations	0.721	0.751	11
	Training and education of safety skills	0.692		
	Conscientiousness	0.792		
	Dependency level of work experience	0.799		
Safety citizenship behaviour	Mutual aid among the workers	0.823	0.813	12
	Relationship between superior and subordinate	0.806		
	Participation in suggestion making	0.773		
	Self-control	0.851		

**Table 8 ijerph-16-01411-t008:** Results of confirmatory factor analysis for questionnaires.

Questionnaire	**χ^2^/df**	RMR	GFI	IFI	CFI	PGFI	RMSEA	AVE
Safety consciousness	1.968	0.051	0.883	0.917	0.923	0.658	0.057	0.612
Safety citizenship behaviour	1.827	0.043	0.912	0.942	0.951	0.679	0.042	0.741
Standard	1–2	<0.05	>0.9	>0.9	>0.9	>0.5	<0.05	>0.5

Note: RMR: Root Mean Square Residual; GFI: Goodness-of-Fit Index; IFI: Incremental Fitness Index; CFI: Comparative Fitness Index; PGFI: Parsimonious Goodness-of-Fit Index; RMSEA: Root Mean Square Error of Approximation; AVE: Average Variance Extracted.

**Table 9 ijerph-16-01411-t009:** Correlation analysis between safety citizenship behaviour and various dimensions of safety consciousness.

Variables	Safety Citizenship Behaviour	Safety Skills Training	Dependency of Experience	Conscientiousness	Familiarity with Safety Regulations
Pearson correlation	1	0.403 **	0.258 **	0.597 **	0.551 **
Significance (double tailed)	0.000	0.000	0.000	0.000	0.000
Case number	382	382	382	382	382

Note: ** At the 0.01 level (double tailed), the correlation is significant.

**Table 10 ijerph-16-01411-t010:** Correlation analysis between safety consciousness and various dimensions of safety citizenship behaviour.

Variables	Safety Consciousness	Mutual Aid	Relationship between Superior and Subordinate	Participation of Suggestion Making	Self-Control
Pearson correlation	1	0.616 **	0.112	0.568 **	0.586 **
Significance (double tailed)	0.000	0.000	0.000	0.000	0.000
Case number	382	382	382	382	382

Note: ** At the 0.01 level (double tailed), the correlation is significant.

**Table 11 ijerph-16-01411-t011:** Pearson correlation coefficient between each dimension.

Variables	Safety Skills Training	Dependency on Experience	Conscientiousness	Familiarity with the Laws and Regulations
Mutual aid	0.427 **	0.390 **	0.582 **	0.455 **
Relationship between superior and subordinate	0.030	0.131 *	0.123 *	0.247 **
Making suggestion	0.385 **	0.413 **	0.504 **	0.479 **
Self-control	0.473 **	0.211 **	0.641 **	0.597 **

Note: ** At the 0.01 level (double tailed), the correlation is significant; * At the 0.01 level (double tailed), the correlation is moderate.

**Table 12 ijerph-16-01411-t012:** Fit criteria of SEM (structural equation model).

Model	**χ^2^/df**	RMR	GFI	IFI	CFI	PGFI	RMSEA
SC and SCB	1.768	0.057	0.851	0.828	0.827	0.658	0.057
Standard	1–2	<0.05	>0.9	>0.9	>0.9	>0.5	<0.05

Note: SC: Safety Consciousness. SCB: Safety Citizenship Behaviour.

**Table 13 ijerph-16-01411-t013:** Path coefficients of SEM.

Path			Estimate	Significant
SCB	<---	SC	0.834	***
experience	<---	SC	0.538	***
conscientiousness	<---	SC	0.867	***
education	<---	SC	0.572	***
regulation	<---	SC	0.616	***
help	<---	SCB	0.835	***
relation	<---	SCB	0.359	
suggestion	<---	SCB	0.764	***
self-control	<---	SCB	0.763	***

Note: *** At the 0.01 level (double tailed), the correlation is significant.

**Table 14 ijerph-16-01411-t014:** Linear regression with multiple elements.

Model	Unstandardised Coefficient		Standardised Coefficient	Col-Linearity Statistics		Significance	*R* ^2^
	B	Standard Deviation	Beta	Tolerance	VIF		
(Constant)	1.436	0.156				0.000	0.681
Safety skills training	0.084	0.035	0.111	0.662	1.510	0.017	
Dependency of experience	−0.029	0.023	−0.060	0.616	1.624	0.214	
Conscientiousness	0.338	0.042	0.423	0.513	1.949	0.000	
Familiarity with the laws and regulations	0.260	0.035	0.329	0.741	1.350	0.000	

Note: VIF: Variance Inflation Factor, which should be less than 10 if there are no col-linearity phenomenon.

**Table 15 ijerph-16-01411-t015:** Results of ANOVA in terms of demographic information.

	Feature	Quadratic Sum	Degree of Freedom	Mean Square	*F*	Significance
Between-column	Gender	0.982	1	0.982	3.142	0.032 (<0.05)
Within-group		118.564	380	0.312		
Between-column	Age	2.782	4	0.696	2.241	0.065
Within-group		116.981	377	0.310		
Between-column	Educational background	3.730	3	1.243	4.051	0.007 (<0.05)
Within-group		116.033	378	0.307		
Between-column	Length of service	1.767	5	0.353	1.126	0.346
Within-group		117.996	376	0.314		
Between-column	Weekly working hours	3.652	4	0.913	2.966	0.047 (<0.05)
Within-group		116.116	377	0.308		

**Table 16 ijerph-16-01411-t016:** Average score of safety citizenship behaviour (SCB) for subgroups with significant differences in SCB.

Feature	Subgroup	Average Score of SCB
Gender	Male	4.18
	Female	3.97
Educational background	Junior middle school or below	3.72
	High school	3.93
	Technical school	4.15
	Undergraduate or above	4.21
Weekly working hours	<40	4.17
	40–45	4.20
	46–50	4.24
	51–55	4.12
	>55	3.96

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
