# Peer review of "Development of Scales to Measure and Analyse the Relationship of Safety Consciousness and Safety Citizenship Behaviour of Construction Workers: An Empirical Study in China"

_ijerph, 2019, doi:10.3390/ijerph16081411_

Reviewer 1 Report

First Impression:

The manuscript is well written and based upon rigorous methods, very well referenced, stimulating, and informative. The authors explored the relationship between safety conscious and safety citizen ship as predictors of safety behaviors. The content will be of great interest to many people and I congratulate the authors for their work.  Construction continues to be a high risk industry and work is needed to save lives and reduce injuries. The authors provide a solid framework for interventions that will positively affect workers behaviors and safety on the job.

Strengths:

The manuscript reads very well with only minor editing needed. The study was well designed and executed. Rigorous methods were used. Tables and figures are valued added. The conclusions are based upon strong findings that extends prior research into construction to underscore the complexity of workplace safety in this dynamic industry. This study identified important factors that could lead to the development of interventions to reduce injuries and fatalities and improve the safe work behaviors, health, wellbeing and quality of life for construction workers.

 Weaknesses:

No real weaknesses seen

Comments and Editorial Feedback:

Page 3, line 1 … project and/or organization.

Page 4, line 27 – start the sentence with Four hundred…

Page 12, Conclusion and Limitation – Authors might consider adding that surveys response were self-reported and may be inaccurate due to recall and response bias.

Page 32 and 33 – The translation of the survey sale indicates a higher level of vocabulary and cognition. The Likert scale is usually a level of agreement with the statements. I just wonder if other words are better choices such:

 1 = highly disagree, 2 = disagree, 3 = neither disagree, 4 =  agree,  5 = highly agree.

Author Response

Dear Reviewer:

          I appreciate your valuable comments for my paper, please kindly refer to the cover letter I uploaded for you in which I have responded the comments one by one. All the revised parts have been highlighted with yellow shading in the manuscript.

           Your's

                                                                                                                          MENG Xiangcheng

Reviewer 2 Report

This study explores the relationship between safety consciousness and safety citizenship behavior of construction workers in China. Safety consciousness and safety citizenship behavior are important concepts in safety management. There is potential in this paper; however, the paper has some problems that need to be addressed before the manuscript can be accepted.

1. The title is highly recommended to be changed as the paper is not just to develop the scales.

2. The item numbers in Table 5 and Table 6 should have one to one correspondence with Appendix A.

3. Figure 1 and Figure 2 are screenshots and should be redrawn clearly.

Author Response

Dear Reviewer:

          I appreciate your valuable comments for my paper, please kindly refer to the cover letter I uploaded for you in which I have responded the comments one by one. All the revised parts have been highlighted with water blue shading in the manuscript.

           Your's

                                                                                                                          MENG Xiangcheng
